

# Sea Surface Temperature over the Bay of Bengal: A key driver for South Asian Summer Monsoon rainfall during past 31 kiloyears

Thamizharasan Sakthivel [1,2], Prosenjit Ghosh[1,2] , Ravi Bhushan[3], Harsh Raj[3], Ankur J Dabhi[3],

Ajay Shivam[3], Senthilnathan D[4]

[1]Centre for Earth Sciences, Indian Institute of Science, Bangalore, India 560012.

[2]Divecha Centre for Climate Change, Indian Institute of Science, Bangalore, India 560012.

[3]Geosciences Division, Physical Research Laboratory, Ahmedabad, India 380009.

[4]Department of Earth Sciences, Pondicherry University, Puducherry, India 605014.

*Correspondence to*: Prosenjit Ghosh (pghosh@iisc.ac.in)

**Abstract:**

Warmer Sea Surface Temperature (SST) in the Bay of Bengal (BoB) is crucial for driving deep atmospheric convection, facilitating low-level south-westerly winds, and enhancing moisture transport, thereby intensifying South Asian Summer Monsoon (SASM) rainfall over South Asia. However, the specific impact of BoB SST on SASM rainfall during the Glacial-Interglacial periods remains poorly understood. In this study, we reconstructed SST and evaporation versus rainfall variability over the past 31 kiloyears by simultaneously analyzing the carbonate clumped isotopes and stable oxygen isotopic composition of surface-dwelling planktic foraminifera *Globigerinoides ruber* from the Central West BoB (CWBoB), a key moisture source region. Additionally, cloud cover index was inferred from the abundance ratio of planktic foraminifera *Globigerina bulloides* to *Neogloboquadrina dutertrei*. Our SST reconstruction reveals an 8°C variability over the past 31 kyr, coinciding with shifts in the *G. bulloides to N. dutertrei* ratio during the Last Glacial period and deglaciation, suggesting SST regulation by variable cloud cover. The increase in SST from the Early Holocene is attributed



to $CO_2$ radiative forcing. The stable oxygen isotope of seawater $\delta^{18}O_{sw}$ strongly aligns with a
proxy record of SASM wind intensity, indicating that changes in wind patterns drive the
variable evaporation versus rainfall dynamics over CWBoB. Furthermore, we examined the
temporal variation in SASM continental runoff and rainfall to the Northern BoB (NBoB) by
assessing changes in $\delta^{18}O_{sw}$ ($\Delta^{18}O_{sw}$), a proxy for Sea Surface Salinity ($\Delta SSS$), between the
NBoB and CWBoB. Our analysis revealed a significant relationship between SASM rainfall
and SST in the CWBoB, indicating a sensitivity of 0.9±0.1 psu drop in $\Delta SSS$ across the NBoB
per 1°C rise in SST. These findings enhance our understanding of the relationship between
CWBoB SST and SASM rainfall, highlighting the intricate dynamics of monsoon variability
and paving the way for improved predictability of SASM rainfall patterns.
**1. Introduction:**
Bay of Bengal (BoB) contributes 45-75% of total moisture to South Asian Summer Monsoon
(SASM) rainfall (Dar and Ghosh, 2016; Yoon and Chen, 2005), which accounts for 78% of
annual rainfall in the region (Parthasarathy et al., 1994). Variability in SASM rainfall
significantly affects economic growth for populations hosted in South Asian countries (Gadgil
and Gadgil, 2006). This highlights the significance of SASM rainfall variability as a crucial yet
uncertain factor in regional climatology. The thermodynamic conditions of seas surrounding
the continent have an important role in the future variability of SASM rainfall (Sein et al.,
2024; Sharma et al., 2023; Dinezio et al., 2020). The optimal sea surface temperatures (SST)
in the BoB, exceeding the range of 26-28°C, act as a threshold for initiating and sustaining the
process of deep atmospheric convection (Shenoi, 2002). This contributes to increased upper-
tropospheric temperatures, driving low-level winds from the south-west direction. These winds
integrate regional moisture, leading to the intensification of SASM rainfall (Goswami, 1987;
Hurley and Boos, 2015; Shenoi, 2002; Samanta et al., 2018). Any decrease in SST over the
BoB during summer is associated with the failure of the SASM (Vecchi and Harrison, 2002).



Even the inaccurate simulation of BoB SST in forecast climate models introduces bias in
SASM rainfall predictions (Samanta et al., 2018). However, despite this thermodynamic
understanding, the relationship between BoB SST and SASM rainfall is disrupted in the
instrumental age record due to the interplay of short-term climate phenomena such as El Niño-
Southern Oscillation, the Indian Ocean Dipole, and the North Atlantic cooling (Goswami et al.,
2022; Saxena and Pandey, 2021). These phenomena influence global oceanic and atmospheric
processes, including the moisture transport process over the Bay of Bengal (Chakraborty and
Singhai, 2021; Borah et al., 2020). The long-time-integrated record reduces interference from
transient climatic phenomena described above (Wang et al., 2017), offering invaluable insight
into the sensitivity of climatological BoB SST to SASM rainfall variability.
The Last Glacial and Holocene periods offer a valuable time window for estimating the role of
BoB SST on regional moisture transport. This time frame appropriately represents a significant
change in atmospheric $CO_2$ concentrations, ranging between 180 and 280 ppm (Bereiter et al.,
2015). Additionally, there is a difference in summer solar insolation values between 30°N and
the equator, varying between 84 W/m$^2$ and 102 W/m$^2$ (Berger, 1992). These variations
contribute to land-sea thermal and pressure contrasts, which in turn affect SASM wind strength
and rainfall distribution (Goswami et al., 1999; Webster, 1994; Rajeev et al., 2012; Evans et
al., 2015). Also, a geographic configuration similar to modern-day one serves as a template for
validating climate model predictions with variable SST conditions globally and over the BoB
region (Tierney et al., 2020; DiNezio et al., 2018). The role of BoB SST in influencing SASM
intensity has been addressed in a limited number of studies. These studies utilize proxies, such
as the stable hydrogen isotopic composition of leaf wax (δD) and the Ba/Ca ratio in planktic
foraminifera *G. ruber* from sediment cores in the Northern BoB (NBoB), to qualitatively
understand SASM rainfall variability (Wang et al., 2022; Weldeab et al., 2022). Here, the
authors compromised with the susceptibility of the isotopic signature preserved in δD of leaf



wax to deduce rainfall amount due to the interplay of factors such as the isotopic signature of
a moisture source, moisture recycling, and isotopic fractionation associated with shifts in
vegetational type (Sachse et al., 2012). The parameter Ba/Ca in *G. ruber* is interpreted as a
proxy for Sea Surface Salinity (SSS), assuming that riverine input is the primary driver for SSS
(Weldeab et al., 2022). However, this assumption is often invalidated by processes such as
evaporation or rainfall occurring within a sea or ocean (Hönisch et al., 2011).
Here, we retrieved a 4.3 m gravity core (MGS17/GC02) at the coordinates 15°19'36" N,
84°54'03" E, at a water depth of 2986m, situated within the Central West Bay of Bengal
(CWBoB) region (Fig. 1a). The sedimentary succession consists of dark grey and grey layer
with variable organic content and their primary constituents are hemipelagic clay with
calcareous foraminifera and nannofossils. The calcareous sediments are authigenic, and their
composition represents a derivative carrying the geochemical signature of average climatology.
We obtained modern-day monthly climatology with a particular interest in Sea Surface Salinity
(SSS) (Reagan et al., 2024) and moisture flux (Trenberth and Fasullo, 2022), which are
important estimates described in the present reconstruction. The observation shows high SSS
coupled with positive moisture flux coinciding with the period of SASM (June to September)
(Fig. 1b). This is aligned with independent observation on the increased freshwater discharge
of the Ganges-Brahmaputra river basin feeding BoB (Fig.1c) (Jana et al., 2015). This implies
the importance of CWBoB as a moisture source for continental rainfall in the modern-day
context, largely similar in paleo-perspective due to minimal change in continental
configuration during the Glacial-Interglacial period. This interpretation is substantiated by a
moisture transport study based on a stable oxygen isotope in rainfall collected over BoB and
South Asia during the period of SASM (Dar and Ghosh, 2016). The influence of riverine
freshwater reaching the location of the present study is found to have minimal events in the
regional ocean general circulation model experimental with and without riverine input (Behara





and Vinayachandran, 2016), making the site a test bed for reconstruction of rainfall over the
oceanic setting.

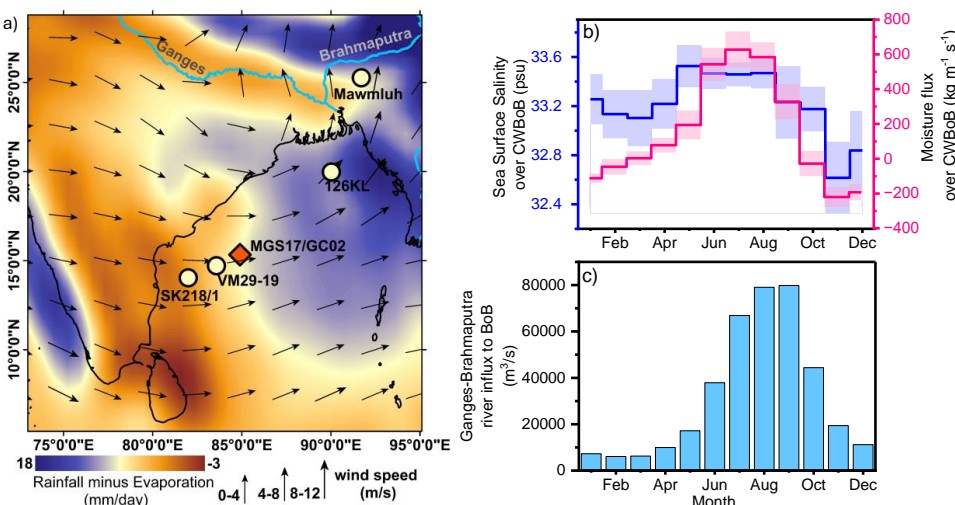

**Figure 1: Site map and climatology.** a) modern-day climatology (1998 – 2019) showing the
distribution of rainfall minus evaporation (mm/day) and wind vectors during the period of
SASM monsoon (June, July, August and September) (Kalnay et al., 1996; Schneider et al.,
2013). The study site, MGS17/GC02 (depicted as a filled orange diamond with a black
boundary), along with other locations (represented by light yellow-filled circles with black
boundaries), are utilized in the discussion; b) Plot of monthly Sea Surface Salinity (Reagan et
al., 2024) and moisture flux (Trenberth and Fasullo, 2022) distribution over the study site
(MGS17/GC02) obtained from world ocean atlas climatology resolved at 4°x4° grid space c)
Long-term average monthly river discharge of Ganges and Brahmaputra to BoB (Jana et al.,

112    2015).


The age-depth model was established utilizing 7 ($^{14}$C) radiocarbon dates obtained from two
species of planktic Foraminifera (*G.ruber* and *G.sacculifer*), revealing sedimentation ages



ranging from 3.6 (at 0.41 m core depth)  to 32.9 (at 4.30 m core depth) kiloyears before present
(kyr BP) (Fig. S1 & Table S1). Bayesian statistics (Bacon) is used to estimate the median age
with uncertainty for the sedimentary layer (Fig. S1) (Blaauw and Christen, 2011).
Here, we conducted carbonate clumped isotope thermometry and stable isotope ($\delta^{18}$O & $\delta^{13}$C)
investigation on planktic foraminifer *G. ruber* across the 18 (+1 replicate) sedimentary depth
with an average time window between each depth of 1.7 kyr, covering the time interval 0-31
kyr BP. The planktic foraminifera *G. ruber*, a surface-dwelling species, thrives at water depths
of 17±20 meters (Lakhani et al., 2022) and is present throughout the year (Maeda et al., 2022;
Guptha et al., 1997), rendering it suitable for reconstruction of surface ocean hydrography
using the geochemical signatures in its carbonate shell. A strong correlation is observed in $\delta^{18}$O
record of *G. ruber* from surface sediment (top 0-1 cm) with estimated $\delta^{18}$O values of *G.ruber*
using collated observation on seawater $\delta^{18}$O values (Table S2) and satellite-based
climatological mean SST values (1991-2020) over BoB during the period of SASM (Fig. S2).
The sedimentary pack of top 1cm represents less than 100 years of accumulated sediments
deduced from the average sedimentation rate (Table S1).  We used species-specific oxygen
isotope thermometry to generate the spatial pattern of $\delta^{18}$O in *G.ruber* carbonate (Mulitza et
al., 2003).
The carbonate clumped isotopes provide a means to reconstruct the equilibrium temperature of
carbonate precipitation independent of the isotopic composition of ambient water (Ghosh et
al., 2006; Tripati et al., 2010; Zaarur et al., 2013; Peral et al., 2018; Daëron and Gray, 2023;
Meinicke et al., 2020). The clumped isotope of *G.ruber*-based SST and $\delta^{18}$O record in *G.ruber*
provide means to reconstruct the equilibrium surface seawater $\delta^{18}O_{sw}$ (details in methods). The
regional hydrological cycle was reconstructed using estimated $\delta^{18}O_{sw}$.



For the first time, we propose a proxy for reconstructing qualitative cloud cover intensity using
the planktic foraminiferal abundance ratio of *Globigerina bulloides* to *Neogloboquadrina*
*dutertrei*, which thrives in different water depths with significant variability in Chlorophyll a
concentration, making it sensitive to the presence of clouds. To quantify cumulative temporal
variations in continental runoff and rainfall over the NBoB, we estimated changes in SSS
($\Delta$SSS) at two different sites with varying distances from the coast. We treated the present site
as open marine with minimal continental runoff compared to the site in proximity to the GBM
river mouth (126-KL) (Fig.1a). Clumped isotope thermometry was used for SST reconstruction
at the present site, while SST at site KL-126 was based on the alkenone unsaturation index
(Kudrass et al., 2001). The temporally distinct SSS for each site was estimated from $\delta^{18}O_{sw}$,
assuming a steady-state Rayleigh oxygen isotope fractionation model with freshwater (rainfall
& continental runoff) and evaporation as input and output fluxes to the BoB surface water
reservoir (details in the methods section).
**2.  Materials and Methods:**
**2.1 Site location and processing of sediment samples:**
The present sampling effort was part of an Exclusive Economic Zone expedition conducted by
the National Centre for Polar and Ocean Research, Ministry of Earth Sciences, India, aboard
RV MGS Sagar (MGS-17) during June-July 2017. The gravity core was retrieved from CW
BoB (15°19'36" N, 84°54'03" E) at the water depth of 2986m (Fig. 1). Our sampling effort
yielded a 4.3m sediment core, which was sub-sampled onboard at 1cm resolution for our
investigation. Each sample was processed onboard, including drying at 50°C in a hot-air oven
for moisture removal and storage in ziplock containers. The samples were transported to the
lab and further processed to remove organics using $H_2O_2$ and sieved to obtain appropriate size
fractions. These fractions were subsequently used for age determination and clumped and
stable isotope analysis.



**2.2 Age - Depth model:**
The age-depth model is proposed based on analysis of mixed species of planktic Foraminifera
(*Globigerinoides ruber* & *Globigerinoides sacculifer*) separated from samples placed at 7
depth intervals (Fig. S1& Table S1). The radiocarbon age determination is conducted at the
Physical Research Laboratory, India, using a 1MV Accelerator Mass Spectrometer. The $^{14}$C-
AMS dates were corrected for the reservoir age offset based on available observations from the
Bay of Bengal (Dutta et al., 2001). This was further adjusted to the calibrated age (before the
present of 1950AD; BP) using calibration software Calib 8.20 (Reimer et al., 2013; Stuiver and
Reimer, 1993). The calibrated dates range from 3580 year BP (41 cm depth) to 32880 year BP
(430 cm depth). In order to establish the sedimentation rate, we assumed the first 1cm core-top
sample as a representation of modern-day (1950 AD). Detailed methodology for sample
analysis and standard reproducibility is provided in the published literature (Bhushan et al.,
2019b, a). We assigned age with uncertainty to each stratum using the Bayesian statistical
(Bacon) method (Fig. S1) (Blaauw and Christen, 2011). Our observation showed variability of
sedimentation rate from 3.86 cm/kyr to 40.35 cm/kyr. A higher sedimentation rate was observed
during the last glacial and late hemispheres, while it was attained at a minimum during
deglaciation and early hemispheres (Table S1). A strong coherence of $\delta^{18}$O variability in *G.*
*ruber* was observed in multiple sites adjacent to our core location (Rashid et al., 2011; Govil
and Divakar Naidu, 2011; Clemens et al., 2021) confirming the proposed age-depth model
(Fig.S3).
**2.3 Stable and clumped isotope analysis in *G.ruber*:**
The present study is based on sediment column samples ranging in age from 0 to 31 kyr BP
with a total of 19 samples (18 samples + 1 replicate). The planktic Foraminifera used here is
*Globigerinoides ruber*, which is most abundant and ubiquitous in the entire succession. The



sediment size fraction of 250–355 μm yielded 8-10 mg of *G. ruber* specimens required for our
analysis. These specimens were initially crushed and treated with 1% $H_2O_2$ and Sodium
Hydroxide buffer to remove any organic matter, further ultrasonicated with methanol to
dislodge clays bound within the foraminiferal skeleton, and finally removed by flotation (Peral
et al., 2018). These samples were dried at 50°C inside a hot air oven to remove moisture prior
to analysis.
The break-seal method is followed for the preparation of $CO_2$ by reacting carbonate powder 8-
10mg with 105% $H_3PO_4$ in an isolated chamber at a constant temperature of 25°C for 18 hours
inside a water bath. The product $CO_2$ is cryogenically extracted and purified for any
contamination using a porapaq-Q GC column held at 25°C (Fosu et al., 2018). The clean $CO_2$
is transferred in an ampule and introduced in a dual inlet peripheral coupled with MAT 253
IRMS configured with a 44-49 mass Faraday cup. The analysis is performed at a major ion
beam intensity of 10V. The working reference gas for dual inlet measurement was sourced from
Linde AG, Munich, Germany, with specifications of 99.999% $CO_2$. It was assigned a value of
$\delta^{13}C$ of -3.92±0.01 ‰VPDB and a $\delta^{18}O$ of 25.58±0.01 ‰ VSMOW, based on repeat analysis
of NBS19 $CO_2$ from carbonate reactions.
Each analytical task consists of a sequence of 5-10 acquisitions, with each acquisition
comprising 10 cycles of sample and working reference $CO_2$ measured alternately. Sample and
working reference $CO_2$ are recovered back following analysis into quartz tube and heated at
1000°C in a muffle furnace for 3 hours to achieve stochastic distribution and defined standard
reference $CO_2$ (heated gas). These heated gases were further processed using the cryogenic
extraction protocol, followed by cleaning using the Porapak-Q column purged with helium.
Multiple heated gases of different bulk compositions are analyzed to correct for non-linearity
and conversion to the heated gas scale (Huntington et al., 2009). Subsequently, these



measurements were transformed into an Absolute Reference Frame (ARF) through a
combination of carbonate reference materials with assigned $\Delta_{47}$ values, including MARJ1,
OMC, ETH1, and ETH3.
The $\delta^{13}C$ and $\delta^{18}O$ values of the sample carbonate are assigned with respect to VPDB, analysing
interlaboratory reference MARJ1 calcite. The $\delta^{13}C$ and $\delta^{18}O$ values of MARJ1 are +1.97‰
and -2.02‰, respectively, established performing experiment with NBS19 as the primary
standard (Ghosh et al., 2005). Routine analysis of MARJ1 carbonate is carried out during the
sample analysis to monitor the long-term variability and performance of the setup. The long-
term reproducibility of $\delta^{13}C$ and $\delta^{18}O$ values of MARJ1 is 0.01‰ and 0.02‰, respectively.
**2.4 Temperature Estimates Based on Clumped Isotope ($\Delta_{47}$) with Uncertainty:**
The most updated calibration (Zaarur et al., 2013) proposed at 25°C acid-carbonate digestion
of inorganic and biogenic (Foraminifera) carbonate precipitation at a known temperature range
of 5°C to 65°C is used for estimation of temperature. The analytical protocol used for this
calibration closely matches the present study. The broad temperature range explored in this
calibration exercise facilitated the appropriate reconstruction of surface ocean temperatures
during the Glacial-Interglacial time. The error propagation due to analytical and calibration
uncertainties is estimated using the suggested algorithm (Huntington et al., 2009).
**2.5 Estimation of $\delta^{18}O_{sw}$ and error propagation:**
The simultaneous measurement of temperature using carbonate clumped ($\Delta_{47}$) isotopes
together with stable oxygen isotopic composition ($\delta^{18}O$) measured in *G.ruber* allows the
reconstruction of $\delta^{18}O_{sw}$ in seawater. We employed the relationship between the $\delta^{18}O$
fractionation between inorganic calcite and water and carbonate precipitation temperature



(Kim and O'Neil, 1997) to estimate the equilibrium $\delta^{18}O_{sw}$ for our sample. This relationship
has been verified for foraminiferal carbonates (Peral et al., 2018; Daëron and Gray, 2023).
The extent of ice volume locked in continental ice sheets also regulates $\delta^{18}O_{sw}$ and is taken into
account for appropriate estimation of $\delta^{18}O_{sw}$. This was accomplished using the equation
(Adkins et al., 2002):
$\delta^{18}O_{sw\text{-}ivc} = \delta^{18}O_{sw} + (SL*0.0083)$
Where $\delta^{18}O_{sw\text{-}ivc}$ is ice volume corrected $\delta^{18}O_{sw,}$ and SL is coral-based sea-level estimates
(Lambeck et al., 2014). The error associated with $\delta^{18}O_{sw\text{-}ivc}$ are estimated by propagating the
errors in SST, and $\delta^{18}O_{G.ruber}$ measurements are given as:
$$\sigma_{\delta^{18}O_{sw\text{-}ivc}}{}^2 = \left(\frac{18030}{SST^2}\sigma_{SST}\right)^2 + (\sigma_{\delta^{18}O_{G.ruber}})^2$$

**2.6 Quantifying the change in SSS between NBoB and CWBoB (ΔSSS):**
The conventional approach to estimating SSS in the Glacial-Interglacial context typically
involves assuming that the modern-day slope and intercept are derived from the correlation
between $\delta^{18}O_{sw}$ and SSS (Govil and Divakar Naidu, 2011). The slope and intercept of this linear
regression equation vary due to differential freshwater fluxes, including runoff and oceanic
rainfall (Singh et al., 2014). This variable behavior of freshwater fluxes contributes to the
overall uncertainty in SSS estimation in the palaeo-reference frame (Mehta et al., 2021). In
order to avoid such complications in SSS estimation, here we used a steady-state Rayleigh
oxygen isotope fractionation model. This model integrates freshwater flux (rainfall and river
runoff) and evaporation as input and output fluxes to the surface ocean reservoir of the Bay of
Bengal (for detailed mathematical derivation, refer to (Singh et al., 2014)). The TraCE-21k
climate model simulation outputs of rainfall and evaporation over the Indian Subcontinent and
the Bay of Bengal are used as flux values in the Rayleigh model (Table S4). The TraCE-21k



climate model simulation reproduced the proxy-based estimates with reasonable satisfaction
(Jalihal et al., 2019, 2020). Assuming an initial composition of the surface ocean reservoir as
0‰ without any freshwater flux, the relationship between SSS and $\delta^{18}O_{sw}$ is expressed as
follows:
$$\frac{SSS}{S_0} = \left[ \frac{\delta^{18}O_{sw} \times (1-\beta)}{(\beta \times \delta^{18}O_{freshwater}) - \varepsilon_{vap/liq}} \right] + 1$$


Where $S_0$ = 34.8 psu represents the mean salinity state of the deep ocean, defined as the initial
salinity reference state. $\beta$ denotes the ratio of fluxes of freshwater to the evaporation. We
assumed Last Glacial Maximum (LGM) fluxes for the period post-LGM which lies between
the time interval of 21-31 kyr BP. The average isotopic composition of freshwater ($\delta^{18}O_{freshwater}$)
is -6.1‰, estimated here by considering different fluxes and their isotopic values from the
literature and weighing them with proportionate contribution (Table S3). The oxygen isotopic
fractionation between vapor and liquid phases during evaporation is represented as $\varepsilon_{vap/liq}$.
Under equilibrium conditions, the SST governs the isotopic composition of vapor (Horita and
Wesolowski, 1994). However, the actual vapor tends to be 2-5‰ heavier than expected at
equilibrium due to the involvement of kinetic processes (Merlivat and Jouzel, 1979). Therefore,
we adopted $\varepsilon_{vap/liq}$ to be 5‰ heavier than the equilibrium state. Applying this method, we
estimated the SSS at the study sites in the CWBoB (MGS17/GC02) and NBoB (KL-126), and
reported the change in SSS ($\Delta$SSS) between these two regions (Table S5). The $\delta^{18}O_{sw}$ for the
KL-126 site was estimated using the same procedure discussed in Section 2.5, employing
alkenone unsaturation index-based SST and the $\delta^{18}O$ of *G. ruber* (Kudrass et al., 2001). The
error propagation associated with $\Delta$SSS estimation is provided in the supplementary text.



**3.  Results and Discussion:**
Below, we describe the stable ($\delta^{18}$O) and clumped isotopic composition in planktic
foraminiferal carbonates (*G.ruber*) at 18 (+1 replicate)- different time intervals across the
sedimentary section encompassing a time interval from 0 to 31 kyr BP located at the Central-
West BoB  (MGS17/GC02). The $\delta^{18}$O values of *G. ruber* varied between -2.0‰ and +0.4‰,
with the heaviest values recorded during the Last Glacial period and the early stages of
deglaciation (15.1-30.9 kyr BP; average  $\delta^{18}$O value of -0.1±0.2‰; n=11) (Table S4).
Progressively, with younging in the sedimentary strata, the $\delta^{18}$O value decreased from +0.2‰
to -1.8‰ denoted by the time interval of late deglaciation (15.1 kyr BP) to the Early Holocene
(9.9 kyr BP) (Table S4). The Holocene (0 to 10.9 kyr BP) is characterized by a lighter average
$\delta^{18}$O value of -1.6 ± 0.5‰ (n=8) (Table S4). The robust consistency in $\delta^{18}$O variability in *G.*
*ruber* in the present study and the data from the adjacent sites (VM29-19(Rashid et al., 2011),
SK218/1(Govil and Divakar Naidu, 2011), and IODP 353 site U1446 (Clemens et al., 2021))
suggest similarity of sea surface temperature and salinity variation in the spatial domain (Fig.
S3). While the $\delta^{18}$O value of planktic foraminiferal carbonate represents habitat temperature
and the composition of seawater $\delta^{18}$O, clumped isotope thermometry provides a unique method
to ascertain temperature for carbonate precipitation in equilibrium without reliance on the
ambient water's $\delta^{18}$O composition (Ghosh et al., 2006; Zaarur et al., 2013; Daëron and Gray,
2023; Meinicke et al., 2020; Peral et al., 2018; Tripati et al., 2010). The clumped isotope ($\Delta_{47}$)
values range from 0.681 to 0.716‰ (Table S4), with heavier values observed during MIS 3 and
the early part of MIS 2, including the LGM, with an average value of 0.704±0.006‰ (n=7)
representing the time interval from 20.9 to 30.9 kyr BP. The $\Delta_{47}$ value decreased from 0.716‰
to 0.683‰ during the later part of deglaciation and Early Holocene (8.7-15.1 kyr BP). The
lowest average $\Delta_{47}$ value of 0.684±0.003 (n=6) was observed during the latest part of the
Holocene (0-8.7 kyr BP). The clumped isotope values are converted into temperature using the



relevant empirical correlation tailored for the foraminiferal carbonates (Zaarur et al., 2013).
Upon estimating the paleo-SST at different time intervals, $\delta^{18}O$ records of carbonates are used
for deriving $\delta^{18}O$ of seawater ($\delta^{18}O_{sw}$), which was corrected for the ice-volume effect (detail
provided in methods).

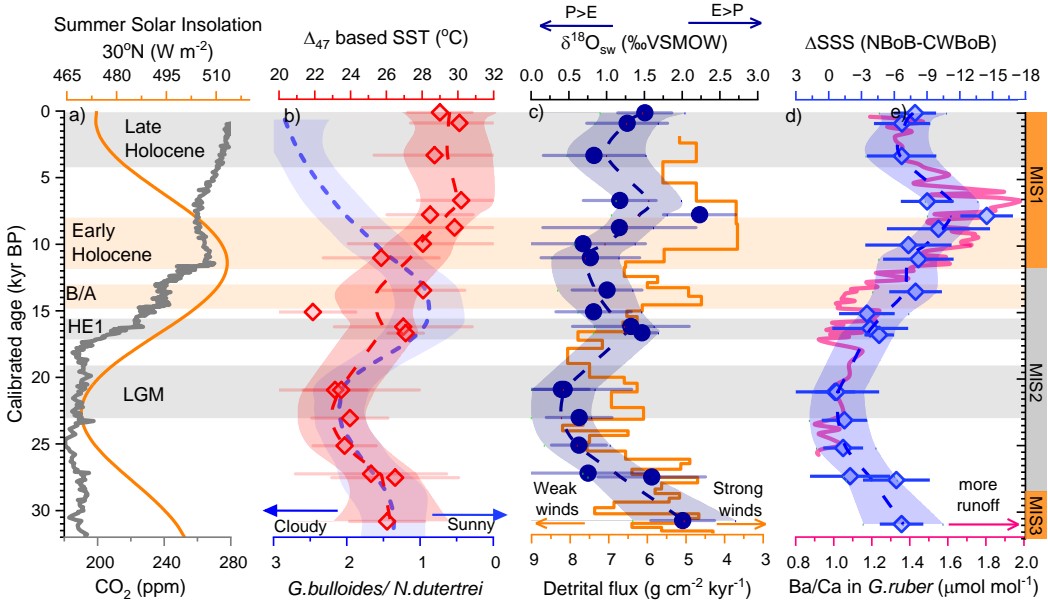


**Fig. 2: Proxy-based records of hydroclimate, Sea Surface Temperature, and its physical**
**drivers over the Bay of Bengal and South Asia for the past 31 kiloyears.** a) Summer (June)
time solar insolation at 30°N (Berger, 1992) and global atmospheric $CO_2$ concentration record
from Antarctica ice core (Bereiter et al., 2015). b) Clumped isotope ($\Delta_{47}$ of carbonate) of
*G.ruber* derived SST from site MGS17/GC02 and cloud cover proxy of *G.bulloides/N.dutertrei*
from site SK218. c) $\delta^{18}O_{sw}$ from the site MGS17/GC02 (CWBoB) compared with SASM wind
proxy of eolian detrital flux (Pourmand et al., 2004). d) Estimated $\Delta SSS$ between NBoB and
CW BoB suggest cumulative freshwater flux to NBoB and Ba/Ca ratio in *G.ruber* record from
NBoB suggesting continental runoff (Weldeab et al., 2022).




### 3.1 Temporal dynamics of SST during the past 31 kilo years:


The SST reconstructed from carbonate clumped isotope ($\Delta_{47}$) data of *G. ruber* over the time
interval of past 31 kyr ranges between 22°C to 30°C (Fig. 2b). Chronologically, the MIS3 and
early phases MIS2, which encompasses a time interval between 27.8-30.9 kyr BP is
characterized by warm average SST ($\pm$1 SD) of 26 $\pm$ 0.7°C (n=3) (Fig. 2b). This is overline by
sediments denoting mid-phase of MIS2 and LGM (20.9-25.2 kyr BP) recording lowest average
SST ($\pm$1 SD) of 24 $\pm$ 0.3°C (n=4). The deglaciation time interval (13.4 - 20.9 kyr BP) recorded
a monotonic increment in the SST values by 5°C (Fig. 2b).  The younger sedimentary sequence
above this represents a time interval of Early Holocene (8.7 - 10.9 kyr BP) which is marked
with average warmer SST ($\pm$1 SD) of 28 $\pm$ 2°C (n = 3) (Fig. 2b). The mid and late Holocene (0
– 7.7 kyr BP) characterized by stable warm SST ($\pm$1 SD) of 29 $\pm$ 0.8°C (n = 5) (Fig. 2b).
The average SST difference between the Late Holocene and the LGM in the present study is
5°C, contrasting with the 3°C difference recorded by Mg/Ca ratio-based SST reconstruction in
*G. ruber* from the region of BoB (Clemens et al., 2021; Rashid et al., 2011; Raza et al., 2017;
Govil and Divakar Naidu, 2011). The discrepancy in SST difference can be explained by the
involvement of non-thermal variables, which include salinity and pH, influencing the Mg/Ca
ratio in *G.ruber*, causing an underestimation of temperature using this approach (Gray and
Evans, 2019).   The organic geochemical proxies-based SST reconstruction, such as the
alkenone unsaturation index ($U^{k'}_{37}$) (Kudrass et al., 2001; Sonzogni et al., 1998), and TEX$_{86}$
(TetraEther index of tetraethers consisting of 86 carbon atoms) (Clemens et al., 2021), recorded
a 2°C difference. These proxies are unaffected by seawater chemistry. However, $U^{k'}_{37}$ faces a
significant limitation in that it becomes insensitive to temperatures above 29°C (Müller et al.,
1998). Additionally, TEX$_{86}$ may primarily reflect subsurface conditions in certain
oceanographic settings (Rommerskirchen et al., 2011). Moreover, both $U^{k'}_{37}$ and TEX$_{86}$ exhibit



seasonal bias when representing SST (Chen et al., 2014; Sonzogni et al., 1998; Wuchter et al.,

343    2006).

The application of the empirical relationship between the abundance of planktic foraminiferal
species with temperature recorded SST shift of 0-2°C over the region of BoB (Cullen, 1981).
The $\delta^{18}O$ record in *G.ruber*-based SST shift with an assumption of constant $\delta^{18}O$ composition
of seawater registered 1°C cooling during LGM compared to Late Holocene (Duplessy, 1982).
The clumped isotope ($\Delta_{47}$) based SST reconstruction in *G. ruber* is independent of non-thermal
variables like pH and salinity beside the isotopic composition of environmental seawater,
making this a superior technique for temperature reconstruction (Tripati et al., 2010; Peral et
al., 2018).
We matched the present SST record over the time interval of the past 31 kyr with global ice
core  $CO_2$ concentration and summer (June) solar insolation at 30°N to understand the role of
internal and external earth system forcing. Our observation explains 54% and 8% variability of
SST with changes in the $CO_2$ concentration and solar insolation, respectively (Fig.S4),
confirming the climate sensitivity analysis from climate model output (Araya-Melo et al.,

357    2015).

The variation in the SST values during the past 31 kyr is explained by combining summer solar
insolation and atmospheric $CO_2$ forcing together, modulating the process of upwelling and
cloud cover associated with convective rainfall. The upwelling process promotes the transfer
of cold subsurface water, while cloud cover associated with convective rainfall determines the
extent of solar radiation reaching the surface water. The upwelling process can be understood
through paleo-productivity proxies, such as biogenic silica records from the BoB, which
indicate weak upwelling from the Last Glacial to the Early Holocene and strong upwelling
during the Mid-Holocene (Liu et al., 2021).



**3.2 Effect of cloud cover on SST:**


The cloud cover plays a pivotal role in regulating Photosynthetically Active Radiation (PAR)
over the ocean, thereby influencing the depth of maximum Chlorophyll a (Chl a) through
mechanisms such as photoacclimation and chlorophyll re-organization (Jyothibabu et al., 2018;
Masuda et al., 2021). Elevated light intensity induces the contraction and aggregation of
chloroplasts within phytoplankton cells, reducing their efficiency in absorbing light and Chl a
content (Kiefer, 1973). Consequently, this triggers the vertical migration of Chl a, leading to
its accumulation in regions characterized by low PAR (Jyothibabu et al., 2018).
Studies conducted during the SASM monsoon period have demonstrated a notable shift in the
water depth of Chl a maxima between cloudy and sunny days, ranging from surface levels to
40 meters and 40-80 meters water depth over the BoB, respectively (Jyothibabu et al., 2018);
identified using onboard observation in the region of BoB. The abundance of planktic
foraminifera is sensitive to Chl a concentrations (Munir et al., 2022; Kuroyanagi and Kawahata,
2004). Specifically, two species, *Globigerina bulloides,* and *Neogloboquadrina dutertrei*,
exhibit maximal thriving conditions at water depths of 0-50 meters and 50-100 meters,
respectively (Tapia et al., 2022) (Fig.S5).
Drawing upon the relationship between cloud cover and the depth of Chl a maxima, as well as
the influence of Chl a on planktic foraminiferal abundance, we propose a proxy for cloud cover
utilizing the ratio of planktic foraminiferal abundance of *G. bulloides* to *N. dutertrei*. A
significant negative correlation is evident between reanalysis data of outgoing longwave
radiation, serving as an index for cloud cover, and the ratio of *G. bulloides* to *N. dutertrei*, as
revealed by sediment trap records from the NBoB (Pearson's r value = -0.61 , p-value = 0.03,
n = 13) and CBoB (Pearson's r value = -0.74 , p-value = 0.004, n = 13) (Guptha et al., 1997)
(Fig. S6). The relatively weak relationship and differential slope observed over NBoB
compared to CBoB may be attributed to the influence of riverine suspended sediments, which





also modulate the relationship between PAR and the depth of Chl a maxima (Jyothibabu et al.,
2018). This framework enables the reconstruction of paleo-cloud cover using temporal
variation in the ratio of *G. bulloides* to *N. dutertrei*. It is important to mention that the upwelling
process also contributes to the shifting of the depth of Chl a maxima (Garg et al., 2024), thereby
contributing to the populational abundance of *G. bulloides* (Prell and Curry, 1981). The SASM
monsoon also generates upwelling along the eastern margin of India (Shetye et al., 1991).
However, the observation showed weaker upwelling at the site of the present study with
confinement near the coast because of the strong stratification (Shetye et al., 1991;
Gopalakrishna and Sastry, 1985).
We adapted the planktic foraminiferal abundance data recorded in the sediment core of the
adjacent site SK218 (Verma et al., 2022) to derive information about the role of cloud cover on
regional SST. Given the weak upwelling activity between Early Holocene and Last Glacial
based on biogenic silica mass accumulation rate (Liu et al., 2021), cloud cover emerged as the
primary factor modulating the ratio of *G. bulloides* to *N. dutertrei*. The high and low ratio
values of *G. bulloides* to *N. dutertrei* during the LGM and the deglaciation to Early Holocene
periods indicate periods of high and low cloud cover, respectively. However, during the middle
Holocene, the high ratio of *G. bulloides* to *N. dutertrei*, coupled with high biogenic silica
content (Liu et al., 2021), suggests that both cloud cover and upwelling were dominant factors.
The temporal correlation observed between the ratio of *G. bulloides* to *N. dutertrei* and changes
in SST (Fig. 2b), as inferred from our $\Delta_{47}$-based temperature reconstructions, suggests that
cloud cover has significantly influenced SST dynamics during the time interval of 16-31 kyr
BP in the CW BoB. From 16 kyr BP to 0 kyr BP, SST in the CWBoB increased despite a rise
in cloud cover (Fig. 2b). This warming trend can be attributed to radiative forcing associated
with elevated atmospheric $CO_2$ levels (Fig. 2a & 2b).



**3.3 Evaporation versus Rainfall over CW BoB regulated by wind:**

SASM is characterised by SW trade wind, which promotes evaporation and release of latent heat over the region of CW BoB (Samanta et al., 2018) and transports moisture to the region of NBoB and South Asia (Dar and Ghosh, 2016; Yoon and Chen, 2005). Here, we took advantage of the estimated $\delta^{18}O_{sw}$ at CW BoB to reconstruct the moisture imbalance due to the process of evaporation and rainfall (Fig. 2c). The strength of SASM wind during this time interval is derived from eolian detrital flux from the northeastern Arabian Sea (Pourmand et al., 2004) (Fig. 2c). The eolian input to this region is transported from the Arabian Peninsula and Persian Gulf during periods of weak SASM winds and strong northwesterly. Conversely, the eolian input is minimal during periods of strong SASM winds (Pourmand et al., 2004).

The $\delta^{18}O_{sw}$ record from CWBoB exhibits a robust temporal correlation with the SASM wind speed proxy (eolian detrital flux) spanning the past 31 kyr (Fig. 2c). Lighter $\delta^{18}O_{sw}$ values observed during the early phase of MIS 2 (20.9 – 27.6 kyr BP; including LGM), the deglaciation to Early Holocene transition (9.9 – 13.4 kyr BP), and the mid to late Holocene transition (3.2 – 6.6 kyr BP) indicate surplus rainfall relative to evaporation over the CW BoB (Fig. 2c). This phenomenon is linked to weakened SASM wind speeds, reducing the quantum of moisture transported to the NBoB and South Asia, thereby exacerbating aridity on the continent (Fig. 2c) (Kudrass et al., 2001; Dutt et al., 2015).

Conversely, heavier $\delta^{18}O_{sw}$ compositions during the MIS 3 to MIS 2 transition (27.6 – 30.9 kyr BP), Heinrich Event 1, and the Early to mid-Holocene transition (6.6 – 9.9 kyr BP) suggest an excess of evaporation compared to rainfall over the CW BoB (Fig. 2c). This trend is associated with intensified SASM winds (Fig. 2c) and increased continental rainfall and river runoff, indicating moisture transport from CWBoB to South Asia (Kudrass et al., 2001; Dutt et al., 2015; Weldeab et al., 2022). In summary, the fluctuation in evaporation over rainfall in the CW

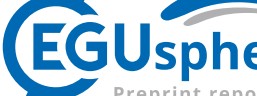

BoB over the past 31 kyr implies a meridional shift in the intensity of convective rainfall clouds
during this period.

**3.4 Moisture transport to the continent in response to SST over CW BoB:**

The moisture transport to South Asia and NBoB is driven by two primary physical factors,
which include evaporated moisture flux from CW BoB together with wind, which propel the
water vapor parcel to reach the coastal region and continental landmass (Dar and Ghosh, 2016;
Yoon and Chen, 2005; Shenoi, 2002; Samanta et al., 2018). The contribution of moisture flux
varies with changes in SST, while the transport is governed by the pressure difference between
ocean and continental landmass (Samanta et al., 2018; Goswami, 1987). The region of CW
BoB during SASM time based on modern-day observation shows the intensification of the
evaporation process with higher SST (Samanta et al., 2018). The sensitivity of this process
varied during Glacial and Inter-glacial time frames with 8°C change in the SST condition over
CWBoB (Fig. 2b). The proportion of this moisture distribution into the NBoB as rainfall and
continental runoff can be understood with an independent record of reconstructed $\delta^{18}O_{sw}$
variability across Glacial and Interglacial time interval. To quantify the cumulative freshwater
influx into the NBoB, we estimated the difference in $\delta^{18}O_{sw}$ ($\Delta^{18}O_{sw}$) between the coastal site
KL-126 in the NBoB (Kudrass et al., 2001) and the present study site MGS17/GC02 in the CW
BoB.  The $\delta^{18}O_{sw}$ over BoB is related to Sea Surface Salinity (SSS), which is modeled using an
oxygen isotope-based Rayleigh steady-state model with freshwater and evaporation as input
and output to the surface seawater reservoir (Singh et al., 2014). The ($\Delta^{18}O_{sw}$) is used for the
estimation of changes in SSS ($\Delta$SSS) between NBoB and CW BoB for the Glacial-Interglacial
time interval, which translates into a relative change in rainfall over South Asia and continental
runoff to NBoB (Fig. 2d & Table S4). The temporal pattern of $\Delta$SSS is in accordance with
Ba/Ca ratio of *G.ruber*-based continental runoff proxy from the region of NBoB, which reflects
SASM rainfall intensity over the continental region (Weldeab et al., 2022) (Fig. 2d).



The higher average ( ± 1SD ) ΔSSS value of -1.4 ± 0.4 psu (n = 4) is observed during the early
phase of MIS 2 (20.9-28 kyr BP), prohibiting moisture transport to the continental region (Fig.
2d & Table S4). This is consistent with our observation of the high ratio of *G. bulloides* to *N.*
*dutertrei* abundances, denoting the presence of convective cloud cover over the region of CW
BoB (Fig. 2b).
The lower average ( ± 1SD ) ΔSSS value of -11.2 ± 2.9 psu (n = 3) is observed during the Early
to Mid Holocene transition  (6.6 to 8.7 kyr BP), corresponding to a period of excess continental
runoff over NBoB (Fig. 2d). The evidence of rainfall variability in the continental region is
discernible in the vegetation cover, as revealed by the temporal pollen records from lake
sediments. These records indicate the presence of open vegetation during periods of high ΔSSS
and mixed tropical deciduous vegetation during periods of low ΔSSS (Quamar and Bera, 2020).
We observed a strong relationship between SST at CW BoB with ΔSSS denoting freshwater
input into the region of NBoB (Fig. 3). This relationship is based on 18-time intervals (+1
replicate) in continuity in the sedimentary record, denoting average time resolution of 1.7 kyr
which discounts interferences from short term climatic processes such as El Niño-Southern
Oscillation, Indian Ocean Dipole, and North Atlantic Oscillation. The observation shows a
sensitivity of moisture transport as evident from a drop in ΔSSS by 0.9±0.1 psu with an
increment of 1°C in the SST record over CW BoB (Fig. 3). Our observation identified SST as
a crucial factor in determining the moisture transport process and rainfall over NBoB and South
Asia.



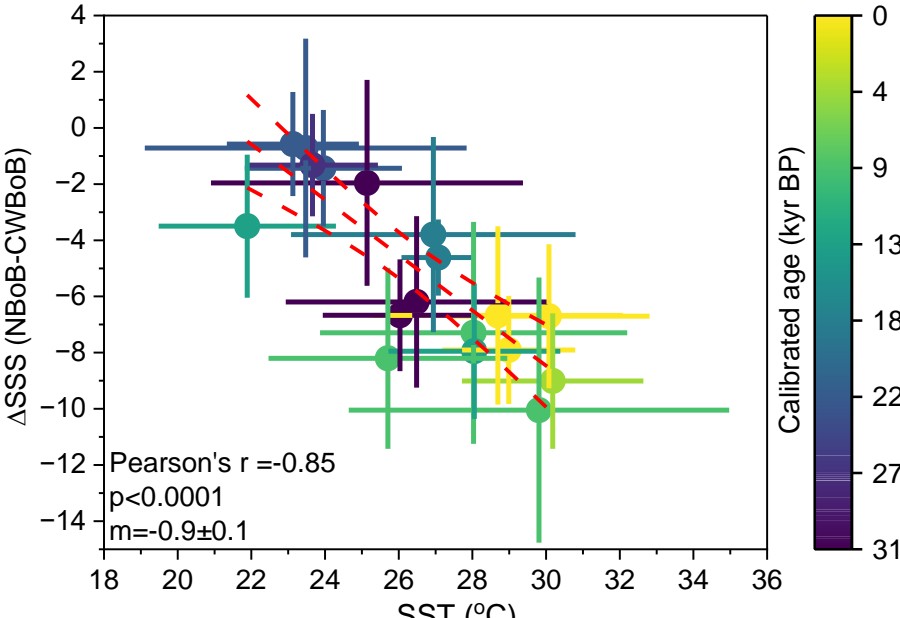

**Fig. 3:** Time-integrated record showing the response of ΔSSS (NBoB-CWBoB) with SST over

CW BoB denotes the intensity of SASM over South Asia and NBoB during the past 31kyr BP.

The color grading provides measures for geological time intervals.

## 4. Conclusion:

This is the first record of SST variability from BoB covering the time period of the past 31 kyr

BP encompassing interglacial and last glacial time interval using clumped isotope thermometry

on *G.ruber* planktic foraminiferal carbonate. The variability in SST is predominantly driven by

global atmospheric $CO_2$ levels, accounting for 54% of the overall signal. Notably, internal

feedback processes involving cloud cover emerge as significant factors in SST modulation.

Furthermore, hydrological feedback mechanisms within the CWBoB are elucidated through



$\delta^{18}O_{sw}$ analysis. A strong correlation is observed between $\delta^{18}O_{sw}$ and SASM wind strength from
31 kyr BP to the present. The present record showed SST over CW BoB regulated the moisture
transport to NBoB and, in turn, contributed to freshwater discharge by rivers together with
rainfall. This is captured in the SSS difference between the coastal sites at NBoB and open
ocean site at CW BoB.
**Author contributions:**
TS and PG conceptualized the study, and PG secured project funding. TS conducted stable and
clumped isotope analysis on foraminifera and interpreted the data. RV, HR, AJD, and AS did
radiocarbon analysis. TS and PG wrote the original draft and finally reviewed and edited by all
authors.
**Competing interests:**
The authors declare that they have no competing interests.
**Acknowledgements:**
TS expresses gratitude to Dr. John Kurian, Dr. M. Ravichandran, the director of NCPOR, and
expedition chief scientist Mr. Bijesh C M for granting the opportunity to participate in the
scientific cruise aboard RV MGS Sagar 17 and for their assistance in collecting sedimentary
core samples for this study. We extend our appreciation to Dr. Partha Sarathi Jena for his
assistance in constructing the Bayesian age (Bacon) model. Additionally, we acknowledge Dr.
Arvind Singh for providing modern-day surface water oxygen isotope data over the Bay of
Bengal. This research benefited from project grants provided by the Department of Science &
Technology, India. TS acknowledges the Council of Scientific and Industrial Research,
Government of India, for financial support through a fellowship for PhD (File No.: 09/079
(2811)/2019-EMR-1), as well as the Divecha Centre for Climate Change, IISc, India, for the
Grantham fellowship.



**Data availability:**

All data needed to evaluate the conclusions in the paper are presented in the Supplementary

Table (S1-S5).

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
