# Peer review of "Sea Surface Temperature over the Bay of Bengal: A key"

_EGUsphere, 2024_

## Author Comment (AC1)

The referee's comments are in italics, and our responses are non-italics.

**Comment 1:** *This manuscript examines the influence of Sea Surface Temperature (SST) in the Bay of Bengal (BoB) on the South Asian Summer Monsoon (SASM) rainfall over the past 31 ka. It is done by analyzing various proxy records, including carbonate-clumped isotopes and stable oxygen isotopic composition of surface-dwelling planktic foraminifera Globigerinoides ruber from the Central West BoB (CWBoB). This work is intriguing as the multiple proxy records provide valuable insights in enhancing our understanding of the dynamics of SASM rainfall. I have the following suggestions and questions.*

1. *To verify the importance of CWBoB as a moisture source for continental rainfall in the modern-day context (lines 92-95), a figure illustrating moisture transport using observational data would be helpful.*

**Response:** Thanks for appreciating our work.

Thanks again! We introduced a revised figure as a modified version of Figure. 1. In the revised illustration, we added a new panel (b), which shows the ERA-Interim moisture flux (as contours) during the SASM for the period 1979 - 2017 together with the 48-hour backward air-mass trajectories information (prior to rainy days) at multiple altitudes (200m, 500m, and 1000m above mean sea level) for the year 2023. This new panel highlights both the significance of the CWBoB as a moisture source and the moisture transport pathways during the SASM. The updated figure and caption are as follows:

"

[Figure]

**Figure 1: Site map and climatology.** a) modern-day climatology (1998 – 2019) showing the distribution of rainfall minus evaporation (mm/day), wind vectors (black arrow), and surface ocean circulation (green dashed arrow) during the period of SASM (June, July, August, and September) (Kalnay et al., 1996; Schneider et al., 2013; Phillips et al., 2021). The study site, MGS17/GC02 (depicted as a filled orange diamond with a black boundary), along with other locations (represented by light yellow-filled circles with black boundaries), which are discussed in the text; b) contours of mass-corrected vertically integrated moisture flux during the SASM for the period of 1979 - 2017 (Trenberth and Fasullo, 2022), superimposed with 48-hour HYSPLIT backward air mass trajectories (Stein et al., 2015) at multiple altitudes (200 m, 500 m, and 1000 m above mean sea level) for the days prior to SASM precipitation events in 2023 at Kolkata. c) Plot of monthly Sea Surface Salinity (Reagan et al., 2024) and moisture flux (Trenberth and Fasullo, 2022) distribution over the study site (MGS17/GC02) obtained from world ocean atlas climatology resolved at 4°x4° grid space (indicated by the red dashed square in panel b); d) Long-term average monthly river discharge of Ganges and Brahmaputra to BoB (Jana et al., 2015)."

**Comment 2:** *2. Since δ¹⁸O is a mixed signal influenced by temperature, isotopic composition of the water source, ocean pH, and other factors, how could the conclusion of "similarity in sea surface temperature and salinity variation across the spatial domain" (lines 289-292) be drawn?*

**Response:**

This query is linked with Fig. S3, which shows the Z-score of the $\delta^{18}O$ in *Globigerinoides ruber* at our study site and an adjacent location. The $\delta^{18}O$ in *G.ruber* depends on temperature and $\delta^{18}O$ of seawater (related to salinity change). The other factors affecting $\delta^{18}O$ in *G.ruber* are vital effect and pH (which the referee has mentioned). Since the size fraction of *G. ruber* used for stable isotope analysis is 250-355 μm for all the sites, we assume that vital factors such as symbiont photosynthesis, foraminiferal respiration, and calcification exert a similar influence on $\delta^{18}O$ across the sites.

There exists no relationship between pH and $\delta^{18}O$ in *G. ruber*. However, Spero et al. (1999) reported a slope of -0.0022‰ per μmol kg⁻¹ of carbonate ion concentration $[CO_3^{2-}]$ effect in *G. ruber* $\delta^{18}O$.

In the revision, we will restate the lines 289 -293 as follows:

"The consistent $\delta^{18}O$ variability observed in *G. ruber* in our study, along with data from adjacent sites VM29-19 (Rashid et al., 2011), SK218/1(Govil and Divakar Naidu, 2011), and IODP 353 site U1446 (Clemens et al., 2021)), suggest similar variations in SST, SSS, and surface water carbonate ion $[CO_3^{2-}]$ concentration in a spatial domain (Fig. S3). During the LGM, the $CO_3^{2-}$ concentration in tropical surface oceans decreased by approximately 70 μmol kg⁻¹ compared to Late Holocene, resulting in a $\delta^{18}O$ shift of -0.154‰ in *G. ruber* (Köhler and Mulitza, 2024; Spero et al., 1999). This effect of $CO_3^{2-}$ ion concentration accounts for about 8-10% of the total $\delta^{18}O$ change observed between the LGM and the Late Holocene, which ranges from 1.5 to 2.0‰, highlighting SST and SSS as major drivers of $\delta^{18}O$ variability (Table S6).".

**Table S6:** Effect of carbonate ion concentration on the $\delta^{18}O$ difference in *G.ruber* between the LGM and Late Holocene, based on a slope of -0.0022‰ per μmol kg⁻¹ (Spero et al., 1999).

| Site Name | $\delta^{18}O$ in *G.ruber* (‰VPDB) | | | | Difference between LGM and the Late Holocene | Carbonate ion effect (%) |
| --- | --- | --- | --- | --- | --- | --- |
| | LGM (19 - 23 kyr BP) | | Late Holocene (0 - 4 kyr BP) | | | |
| | Mean | 1SD | Mean | 1SD | | |
| MGS17/GC02 | -0.2 | 0.0 | -1.8 | 0.3 | 1.5 | -10 |
| VM29-19 | -1.2 | 0.2 | -2.9 | 0.2 | 1.7 | -9 |
| SK218/1 | -0.9 | 0.2 | -2.8 | 0.3 | 1.9 | -8 |
| IODP353 site U1446 | -0.9 | 0.0 | -2.9 | 0.1 | 2.0 | -8 |

**Comment 3:** *3. As different proxies are used at different sediment cores to reconstruct temperature, to what extent do the uncertainties associated with these different proxies influence the final calculation of δ¹⁸Osw? More discussion should be added.*

**Response:**

Thanks! We will modify the discussion with the following changes to incorporate uncertainties in $\delta^{18}O_{sw}$ estimates starting from line 348:

"Instrumental and calibration uncertainties for various temperature proxies are generally: 2°C for $\Delta_{47}$ thermometry (Zaarur et al., 2013), 1.4–1.8°C for the Mg/Ca ratio in *G.ruber* (Rosenthal et al., 2022), 1.5°C for $U^{k'}_{37}$ (Müller et al., 1998), 4–5°C for $TEX_{86}$ (Kim et al., 2010), 0.7°C for planktic foraminiferal transfer functions (Cullen, 1981), and 0.9°C for $\delta^{18}O$ thermometry (Mulitza et al., 2003). For the SST value of 25°C, the propagated uncertainties in $\delta^{18}O_{sw}$ estimates for these proxies are 0.4‰ for $\Delta_{47}$, 0.3–0.4‰ for Mg/Ca, 0.3‰ for $U^{k'}_{37}$, 0.8–1.0‰ for $TEX_{86}$, and 0.2‰ for both the planktic foraminiferal transfer function and $\delta^{18}O$ thermometry.

Despite the relatively high uncertainty associated with $\Delta_{47}$-based SST reconstructions and propagated $\delta^{18}O_{sw}$, this method is advantageous due to its independence from non-thermal variables such as pH, salinity, and the isotopic composition of seawater (Tripati et al., 2010; Peral et al., 2018). Furthermore, it does not have temperature range limitations for reconstruction (Zaarur et al., 2013). These characteristics make $\Delta_{47}$-based SST an effective method for temperature reconstruction."

**Comment 4:** *4. In Section 3.2, cloud cover reconstruction, which is important and challenging, is addressed. While the authors have made an attempt, I remain unconvinced due to the numerous assumptions and large uncertainties involved. I would suggest to first examine the relationship between cloud cover and sea surface temperature (SST) using observational data.*

**Response:**

Thanks for appreciating the effort in our attempt to reconstruct cloud cover using the planktic foraminifera abundance ratio in Section 3.2. Your concern regarding the assumptions and uncertainties in our approach is highly logical. In response, we will discuss issues such as the driver of ecological niche and foraminiferal dissolution.

In the revision, we will modify the lines 378–381, as follows:

"The premise behind our analysis on the abundance of two planktic foraminiferal species, *Globigerina bulloides* and *Neogloboquadrina dutertrei*, which exhibit optimal thriving conditions at water depths of 0-50 meters and 50-100 meters, respectively, based on multi-plankton net samples from the northern Indian Ocean (Tapia et al., 2022) (Fig. S5). The

oxygen isotope-based apparent calcification depths of these species further indicate shallow water habitat for *G. bulloides* and deeper water habitat for *N. dutertrei* (Stainbank et al., 2019). Given that the salinity and temperature of both shallow and deeper waters in the BoB align with these species' ecological niches, the Chl a concentrations regulates their populations (Lombard et al., 2009; Bijma et al., 1990; Munir et al., 2022; Kuroyanagi and Kawahata, 2004; Maeda et al., 2022)."

We will also revise the discussion between line no. 400 and 402 as follows:

"We have adapted planktic foraminiferal abundance data from the sediment core of the adjacent site SK218 (Verma et al., 2022) to assess the impact of cloud cover on regional SST. The dissolution of calcareous tests may affect the foraminiferal count, potentially altering the original microfaunal population signature of past environmental conditions.

$$\text{Dissolution index} = 10 \times \frac{\text{Planktic foraminifera resistant to dissolution}}{\text{Planktic foraminifera susceptible to dissolution}}$$

The observed dissolution index is below 5 for most of the samples (Verma et al., 2022), suggesting minimal effect from this process (Berger, 1973). In our reconstruction, we assume the habitat depths of *G. bulloides* and *N. dutertrei* have remained constant over the past 31 kyr."

Additionally, we will move the supplementary figure (Fig. S6, showing the calibration between *G. bulloides*/*N. dutertrei* and OLR from two sediment trap sites in the Bay of Bengal) to the main text. An additional figure will be included as a supplementary figure to illustrate the relationship between OLR and cloud cover fraction over the BoB.

[Figure]

**Fig. S6 (as Fig. 3 in revision) : Relationship between the *G. bulloides* to *N. dutertrei* ratio and cloud cover index deduced from Outgoing Longwave Radiation (OLR) over the BoB.** (a) We presented sediment trap data from NBBT03, located in the NBoB, which was programmed to collect 13 successive samples, each spanning a duration of 27 days, between November 16, 1988, and October 6, 1989. Planktic foraminiferal counts were conducted on samples collected at 2 depths, 967m and 1498m, with a size fraction ranging between 150μm to 500μm(Guptha et al., 1997). (b) Sediment trap CBBT03 was deployed over the region CWBoB, which was operational and coinciding with NBBT03. Planktic foraminiferal counts were conducted on samples collected from one depth at 950m(Guptha et al., 1997). Interpolated monthly OLR values were obtained for both sites for the period of observations at NBBT03 and CBBT03, and a 4°x4° grid was designed with each site as the focal point (Liebmann and Smith, 1996).

[Figure]

**Fig. S6 (new):** Map showing cloud fraction derived from cloud mask data (count of lowest two clear sky confidence levels, cloudy, and probably cloudy/total count) (in panel a) (Platnick et al., 2015) alongside OLR (W/m²) (in panel b) (AIRS, 2013) from 2002 to 2016 over the BoB, including their correlation (in panel c).

In response to the referee's suggestion, we will propose to incorporate an analysis of the relationship between cloud cover and SST using observational data to provide a robust validation of the role of cloud cover on SST (included as a supplementary figure below), and we will reference this in Section 3.2.

[Figure]

**Fig. S7 (new):** Correlation between the 11-year running mean of annual in-situ SST anomalies and standardized low cloud cover anomalies over the BoB from 1966 to 1993 (Rajeevan et al., 2000).

**Comment 5:** *5. Temperature is a crucial factor in shaping the moisture balance. This aspect should be discussed in Sect. 3.3.*

**Response:**

Thanks for the suggestion. We agree that SST also plays a key role in shaping the moisture balance. In response, we will expand Section 3.3 to include a discussion on how temperature influences moisture flux, focusing on how SST affects the atmosphere's ability to hold and transport moisture. The discussion will be added starting from line 438 as follows:

"SST, alongside winds, plays a crucial role in moisture flux by regulating the temperature of the overlying air. As air warms, its capacity to hold moisture increases at the Clausius-Clapeyron rate, approximately 7% per °C (Boer, 1993). During the mid-phase of MIS2 and the LGM (20.9–25.2 kyr BP), colder SSTs weakened atmospheric convection, reducing moisture transport to the NBoB and South Asia while enhancing localized rainfall over the CWBoB (Fig. 2b & 2c). The subsequent 5°C rise in SST during deglaciation intensified evaporation over CWBoB, promoting atmospheric convection and increasing moisture transport to NBoB and South Asia (Fig. 2b & 2c)."

**Comment 6:** *6. The uncertainty of Age-Depth model and related discussion should be added in Sect. 2.2.*

**Response:**

We will add a discussion regarding the uncertainty in the age-depth model from line 176 as follows:

"To get a continuous estimate of age as a function depth in the sediment core, we used the Bayesian statistical method (Bacon) to perform interpolation between depths which has calibrated radiocarbon ages and construct uncertainty estimates that account for possible sedimentation rate changes throughout the core (Blaauw and Christen, 2011). The average 95% confidence interval width is 1.6 kyr, making ability to resolve millennial-scale proxy record variations."

**Comment 7:** *7. Could you clarify what are "late hemispheres" and "early hemispheres" (lines 179-180)?*

**Response:** Thanks for pointing this out. It was indeed an error. In the revised version, we will correct it to "Late Holocene" and "Early Holocene".

**Comment 8:** *8. Please provide the exact duration for drying at 50 ℃ (line 192).*

**Response:** The samples were dried in a hot air oven at 50°C for at least 48 hours to eliminate moisture before analysis.

We will include this information in the revised version as follows: "These samples were oven-dried at 50°C for a minimum duration of 48 hours to ensure complete moisture removal prior to analysis."

**Comment 9:** *9. Authors are encouraged to report their clumped data using the I-CDES system (doi.org/10.1029/2020GC009588), which allows for the comparison of clumped measurements from different laboratories and at various reaction temperatures (line 212).*

**Response:**

Thank you for the suggestion. We agree that reporting clumped isotope ($\Delta_{47}$) data using the I-CDES system improves inter-laboratory comparability. In the revised version, we will include the I-CDES values alongside the ARF scale values in Table S4. However, we will use the ARF scale $\Delta_{47}$ values for temperature estimates to reduce uncertainties related to analytical methodology (acid-fractionation corrections and the acid-carbonate reaction method).

The following sentences will be added after line 212:

"Additionally, the $\Delta_{47}$ values are reported on the I-CDES scale, calibrated against published reference carbonate values for ETH-1 and ETH-3 (Bernasconi et al., 2021). To minimize uncertainties arising from factors such as acid-fractionation, which is scaled at 25°C rather than 90°C (Pramanik et al., 2020), and from variations in the acid-carbonate reaction and extraction methods (Banerjee and Ghosh, 2023), the temperature estimates in this study are based on ARF scale $\Delta_{47}$ values."

**Comment 10:** *10. The equation of temperature estimation used should be provided in Sect. 2.4.*

**Response:**

The following $\Delta_{47}$-Temperature (in kelvin) equation in the ARF scale (Zaarur et al., 2013) will be added to section 2.4.

$$\Delta_{47} = \frac{(0.0555 \pm 0.0027) \times 10^6}{T^2} + (0.0780 \pm 0.0298); \ R^2 = 0.93$$

**Comment 11:** *11. When reporting temperatures, please ensure that the number of decimals is consistent between temperature values and their uncertainties throughout the manuscript.*

**Response:** Thanks. We will ensure consistency in the number of decimal places between temperature values and their associated uncertainties throughout the manuscript in the revised version.

**Comment 12:** *Some technical suggestions:*

*Lines 81-82, 157: "15°19'36" N, 84°54'03" E" -> "15°19'36" N, 84°54'03"E";*
*"2986m" -> "2986 m".*

**Response:** Thanks! We will update this in the revised manuscript.

**Comment 13:** *Line 114: for radiocarbon, full name first and then abbreviation.*

**Response:** Thanks! We will update this in the revised manuscript.

**Comment 14:** *Line 146: Is 126-KL a site name? Please keep it consistent elsewhere in the manuscript (legend in Fig. 1a is 126KL, without dash in between; Line 147 is KL-126).*

**Response:** We will cite it as 126KL.

**Comment 15:** *Line 157: Repeated information.*

**Response:** Thanks! We will remove it in the revised version.

**Comment 16:** *Line 197: please check "porapaq-Q".*

**Response:** Thanks! It will be revised as Porapak Q.

**Comment 17:** *Line 214: "ETH1" -> "ETH-1"; "ETH3" -> "ETH-3". Please check the original paper: doi.org/10.1029/2017GC007385.*

**Response:** Thanks! We will revise it.

**Reference:**

AIRS: AIRS/Aqua L3 Daily Standard Physical Retrieval (AIRS+AMSU) 1 degree x 1 degree V006, Greenbelt, MD, USA, Goddard Earth Sciences Data and Information Services Center (GES DISC), Accessed: 05/09/2024, https://doi.org/10.5067/Aqua/AIRS/DATA301, 2013.

Banerjee, S. and Ghosh, P.: A correction scheme for calcium carbonate clumped isotope (Δ47) thermometric equation depending on sample preparation technique, Appl. Geochemistry, 158, https://doi.org/10.1016/j.apgeochem.2023.105809, 2023.

Berger, W. H.: Deep-sea carbonates: Pleistocene dissolution cycles, J. Foraminifer. Res., 3, 187–195, https://doi.org/https://doi.org/10.2113/gsjfr.3.4.187, 1973.

Berggren, W. A.: Carbon cycling in the glacial ocean: Constraints on the ocean's role in global change, 80–82 pp., https://doi.org/10.1016/0012-8252(95)90055-1, 1995.

Bernasconi, S. M., Daëron, M., Bergmann, K. D., Bonifacie, M., Meckler, A. N., Affek, H. P., Anderson, N., Bajnai, D., Barkan, E., Beverly, E., Blamart, D., Burgener, L., Calmels, D.,

Chaduteau, C., Clog, M., Davidheiser-Kroll, B., Davies, A., Dux, F., Eiler, J., Elliott, B., Fetrow, A. C., Fiebig, J., Goldberg, S., Hermoso, M., Huntington, K. W., Hyland, E., Ingalls, M., Jaggi, M., John, C. M., Jost, A. B., Katz, S., Kelson, J., Kluge, T., Kocken, I. J., Laskar, A., Leutert, T. J., Liang, D., Lucarelli, J., Mackey, T. J., Mangenot, X., Meinicke, N., Modestou, S. E., Müller, I. A., Murray, S., Neary, A., Packard, N., Passey, B. H., Pelletier, E., Petersen, S., Piasecki, A., Schauer, A., Snell, K. E., Swart, P. K., Tripati, A., Upadhyay, D., Vennemann, T., Winkelstern, I., Yarian, D., Yoshida, N., Zhang, N., and Ziegler, M.: InterCarb: A Community Effort to Improve Interlaboratory Standardization of the Carbonate Clumped Isotope Thermometer Using Carbonate Standards, Geochemistry, Geophys. Geosystems, 22, 1–25, https://doi.org/10.1029/2020GC009588, 2021.

Bijma, J., Faber, W. W., and Hemleben, C.: Temperature and salinity limits for growth and survival of some planktonic foraminifers in laboratory cultures, J. Foraminifer. Res., 20, 95–116, https://doi.org/https://doi.org/10.2113/gsjfr.20.2.95, 1990.

Blaauw, M. and Christen, J. A.: Flexible paleoclimate age-depth models using an autoregressive gamma process, Bayesian Anal., 6, 457–474, https://doi.org/10.1214/11-ba618, 2011.

Boer, G. J.: Climate change and the regulation of the surface moisture and energy budgets, Clim. Dyn., 8, 225–239, https://doi.org/10.1007/BF00198617, 1993.

Clemens, S. C., Yamamoto, M., Thirumalai, K., Giosan, L., Richey, J. N., Nilsson-Kerr, K., Rosenthal, Y., Anand, P., and McGrath, S. M.: Remote and local drivers of pleistocene South Asian summer monsoon precipitation: A test for future predictions, Sci. Adv., 7, 1–16, https://doi.org/10.1126/sciadv.abg3848, 2021.

Cullen, J. L.: Microfossil evidence for changing salinity patterns in the bay of Bengal over the last 20 000 years, Palaeogeogr. Palaeoclimatol. Palaeoecol., https://doi.org/10.1016/0031-0182(81)90101-2, 1981.

Govil, P. and Divakar Naidu, P.: Variations of Indian monsoon precipitation during the last 32kyr reflected in the surface hydrography of the Western Bay of Bengal, Quat. Sci. Rev., 30, 3871–3879, https://doi.org/10.1016/j.quascirev.2011.10.004, 2011.

Guptha, M. V. S., Curry, W. B., Ittekkot, V., and Muralinath, A. S.: Seasonal variation in the flux of planktic Foraminifera; sediment trap results from the Bay of Bengal, northern Indian Ocean, J. Foraminifer. Res., https://doi.org/10.2113/gsjfr.27.1.5, 1997.

Jana, S., Gangopadhyay, A., and Chakraborty, A.: Impact of seasonal river input on the Bay of Bengal simulation, Cont. Shelf Res., https://doi.org/10.1016/j.csr.2015.05.001, 2015.

Kalnay, E., Kanamitsu, M., Kistler, R., Collins, W., Deaven, D., Gandin, L., Iredell, M., Saha, S., White, G., Woollen, J., Zhu, Y., Chelliah, M., Ebisuzaki, W., Higgins, W., Janowiak, J., Mo, K. C., Ropelewski, C., Wang, J., Leetmaa, A., Reynolds, R., Jenne, R., and Joseph, D.: The NCEP/NCAR 40-year reanalysis project, Bull. Amer. Meteor. Soc., 77, 437–470, 1996.

Kim, J. H., van der Meer, J., Schouten, S., Helmke, P., Willmott, V., Sangiorgi, F., Koç, N., Hopmans, E. C., and Damsté, J. S. S.: New indices and calibrations derived from the distribution of crenarchaeal isoprenoid tetraether lipids: Implications for past sea surface temperature reconstructions, Geochim. Cosmochim. Acta, 74, 4639–4654, https://doi.org/10.1016/j.gca.2010.05.027, 2010.

Köhler, P. and Mulitza, S.: No detectable influence of the carbonate ion effect on changes in stable carbon isotope ratios ($\delta 13C$) of shallow dwelling planktic foraminifera over the past

160 kyr, Clim. Past, 20, 991–1015, https://doi.org/10.5194/cp-20-991-2024, 2024.

Kuroyanagi, A. and Kawahata, H.: Vertical distribution of living planktonic foraminifera in the seas around Japan, Mar. Micropaleontol., 53, 173–196, https://doi.org/10.1016/j.marmicro.2004.06.001, 2004.

Liebmann and Smith: NOAA Interpolated Outgoing Longwave Radiation (OLR) data provided by the NOAA PSL, Boulder, Colorado, USA, from their website at https://psl.noaa.gov, Bull. Am. Meteorol., 77, 1275–1277, 1996.

Lombard, F., Labeyrie, L., Michel, E., Spero, H. J., and Lea, D. W.: Modelling the temperature dependent growth rates of planktic foraminifera, Mar. Micropaleontol., 70, 1–7, https://doi.org/10.1016/j.marmicro.2008.09.004, 2009.

Maeda, A., Kuroyanagi, A., Iguchi, A., Gaye, B., Rixen, T., Nishi, H., and Kawahata, H.: Seasonal variation of fluxes of planktic foraminiferal tests collected by a time-series sediment trap in the central Bay of Bengal during three different years, Deep. Res. Part I Oceanogr. Res. Pap., 183, 103718, https://doi.org/10.1016/j.dsr.2022.103718, 2022.

Mulitza, S., Boltovskoy, D., Donner, B., Meggers, H., Paul, A., and Wefer, G.: Temperature: δ18O relationships of planktonic foraminifera collected from surface waters, Palaeogeogr. Palaeoclimatol. Palaeoecol., https://doi.org/10.1016/S0031-0182(03)00633-3, 2003.

Müller, P. J., Kirst, G., Ruhland, G., Von Storch, I., and Rosell-Melé, A.: Calibration of the alkenone paleotemperature index U37K based on core-tops from the eastern South Atlantic and the global ocean (60°N-60°S), Geochim. Cosmochim. Acta, 62, 1757–1772, https://doi.org/10.1016/S0016-7037(98)00097-0, 1998.

Munir, S., Sun, J., Morton, S. L., Zhang, X., and Ding, C.: Horizontal Distribution and Carbon Biomass of Planktonic Foraminifera in the Eastern Indian Ocean, 14, 1–15, https://doi.org/10.3390/w14132048, 2022.

Peral, M., Daëron, M., Blamart, D., Bassinot, F., Dewilde, F., Smialkowski, N., Isguder, G., Bonnin, J., Jorissen, F., Kissel, C., Michel, E., Vázquez Riveiros, N., and Waelbroeck, C.: Updated calibration of the clumped isotope thermometer in planktonic and benthic foraminifera, Geochim. Cosmochim. Acta, https://doi.org/10.1016/j.gca.2018.07.016, 2018.

Phillips, H. E., Tandon, A., Furue, R., Hood, R., Ummenhofer, C. C., Benthuysen, J. A., Menezes, V., Hu, S., Webber, B., Sanchez-Franks, A., Cherian, D., Shroyer, E., Feng, M., Wijesekera, H., Chatterjee, A., Yu, L., Hermes, J., Murtugudde, R., Tozuka, T., Su, D., Singh, A., Centurioni, L., Prakash, S., and Wiggert, J.: Progress in understanding of Indian Ocean circulation, variability, air-sea exchange, and impacts on biogeochemistry, Ocean Sci., 17, 1677–1751, https://doi.org/10.5194/os-17-1677-2021, 2021.

Platnick, S., Hubanks, P., Meyer, K., and King, M. D.: MODIS Atmosphere L3 Monthly Product (08_L3)., NASA MODIS Adapt. Process. Syst. Goddard Sp. Flight Cent., https://doi.org/http://modaps.nascom.nasa.gov/services/about/products/c6/MOD08_M3.html, 2015.

Pramanik, C., Chatterjee, S., Fosu, B. R., and Ghosh, P.: Isotopic fractionation during acid digestion of calcite: A combined ab initio quantum chemical simulation and experimental study, Rapid Commun. Mass Spectrom., 34, 1–12, https://doi.org/10.1002/rcm.8790, 2020.

Rajeevan, M., De, U. S., and Prasad, R. K.: Decadal variation of sea surface temperatures, cloudiness and monsoon depressions in the north Indian ocean., Curr. Sci., 79, 283–285, 2000.

Rashid, H., England, E., Thompson, L., and Polyak, L.: Late glacial to holocene indian summer monsoon variability based upon sediment records taken from the bay of Bengal, Terr. Atmos. Ocean. Sci., https://doi.org/10.3319/TAO.2010.09.17.02(TibXS), 2011.

Reagan, J. R., Boyer, T. P., García, H. E., Locarnini, R. A., Baranova, O. K., Bouchard, C., Cross, S. L., Mishonov, A. V., Paver, C. R., Seidov, D., Wang, Z., and Dukhovskoy, D.: World Ocean Atlas 2023., NOAA Natl. Centers Environ. Information.https//www.ncei.noaa.gov/archive/accession/0270533. Accessed [25/04/2024], 2024.

Rosenthal, Y., Bova, S., and Zhou, X.: A User Guide for Choosing Planktic Foraminiferal Mg/Ca-Temperature Calibrations, Paleoceanogr. Paleoclimatology, 37, https://doi.org/10.1029/2022PA004413, 2022.

Schneider, D. P., Deser, C., J. Fasullo, and Trenberth, K. E.: Climate Data Guide Spurs Discovery and Understanding., Eos Trans. AGU, 94, 121–122, https://doi.org/https://doi.org/10.1002/2013eo130001, 2013.

Spero, H. J., Bijma, J., Lea, D. W., and Russell, A. D.: Deconvolving Glacial Ocean Carbonate Chemistry from the Planktonic Foraminifera Carbon Isotope Record, in: Reconstructing Ocean History, Springer US, 329–342, 1999.

Stainbank, S., Kroon, D., Rüggeberg, A., Raddatz, J., De Leau, E. S., Zhang, M., and Spezzaferri, S.: Controls on planktonic foraminifera apparent calcification depths for the northern equatorial Indian Ocean, 1–34 pp., https://doi.org/10.1371/journal.pone.0222299, 2019.

Stein, A. F., Draxler, R. ., Rolph, G. D., Stunder, B. J. B., Cohen, M. D., and Ngan, F.: NOAA's HYSPLIT atmospheric transport and dispersion modeling system, Bull. Amer. Meteor. Soc., 96, 2059–2077, https://doi.org/http://dx.doi.org/10.1175/BAMS-D-14-00110.1, 2015.

Tapia, R., Ho, S. L., Wang, H. Y., Groeneveld, J., and Mohtadi, M.: Contrasting vertical distributions of recent planktic foraminifera off Indonesia during the southeast monsoon: implications for paleoceanographic reconstructions, 19, 3185–3208, https://doi.org/10.5194/bg-19-3185-2022, 2022.

Trenberth, K. and Fasullo, J.: "The Climate Data Guide: ERA-Interim: derived components." Retrieved from https://climatedataguide.ucar.edu/climate-data/era-interim-derived-components on 2024-03-30., Natl. Cent. Atmos. Res. Staff (Eds). Last Modif. 2022-11-07, 2022.

Tripati, A. K., Eagle, R. A., Thiagarajan, N., Gagnon, A. C., Bauch, H., Halloran, P. R., and Eiler, J. M.: 13C-18O isotope signatures and "clumped isotope" thermometry in foraminifera and coccoliths, Geochim. Cosmochim. Acta, 74, 5697–5717, https://doi.org/10.1016/j.gca.2010.07.006, 2010.

Verma, K., Singh, A. D., Singh, P., Singh, H., Satpathy, R. K., Uddandam, P. R., and Naidu, P. D.: Monsoon-related changes in surface hydrography and productivity in the Bay of Bengal over the last 45 kyr BP, Palaeogeogr. Palaeoclimatol. Palaeoecol., 589, 110844, https://doi.org/10.1016/j.palaeo.2022.110844, 2022.

Zaarur, S., Affek, H. P., and Brandon, M. T.: A revised calibration of the clumped isotope thermometer, Earth Planet. Sci. Lett., https://doi.org/10.1016/j.epsl.2013.07.026, 2013.

---

## Author Comment (AC2)

The referee's comments are in italics, and our responses are non-italics.

**Comment 1:** *Review on "Sea Surface Temperature over the Bay of Bengal: A key driver for South Asian Summer Monsoon rainfall during past 31 kiloyears" by Sakthivel et al.,*

*This topic is of scientific relevance, identification paleo-hydroclimates and a key driving forcing of South Asian Summer Monsoon from MIS 3 by using clumped isotopes and stable isotope records of Central West Bay of Bengal (CWBoB). I would suggest to reject this MS because the authors have some big problems on their data and interpretations. To sum up, these data cannot support the main points of the MS. Here I draw my comments as follows.*

**Response:** We appreciate the time and effort the referee has dedicated to reviewing our manuscript. We value their feedback and would like to address their comments constructively.

However, the comment regarding the scientific relevance of our manuscript may require further clarification. Specifically, we are uncertain about the nature of the concerns raised about the data and its interpretation. Our manuscript focuses on the role of SST over the CWBoB in regulating moisture transport over the NBoB and South Asia, as detailed in Section 3.4 and illustrated in Figure 3.

Given this context, we would appreciate additional details from the referee to understand their concerns better and to address them effectively.

**Comment 2:** *1. In the Results and Discussion 3.2, the authors mentioned that the relationship existing between cloud cover and the depth of Chl a maxima, as well as the influence of Chl a on planktic foraminiferal abundance (Line 382-384). Cloud cover index was inferred from the abundance ratio of planktic foraminifera Globigerina bulloides to Neogloboquadrina dutertrei. The authors further suggest that internal feedback processes involving cloud cover index serve as significant factors in SST modulation in the BoB. However, the authors met some fundamental problems. Salinity, nutrient level, prey abundance, turbidity and illumination also affect their diversity, abundance and distribution locally. Besides, depth habitats of cold water dweller bulloides and N. dutertrei live at thermocline. Thus, the relationship between Chl a and planktic foraminifera cannot be applicable to all species. The authors ignore the important component - ocean where planktic foraminifera live. The abundance ratio of planktic foraminifera G. bulloides to N. dutertrei should be reflected changes in ocean hydrography, not only for cloud cover itself. In the MS, cloud cover index and related interpretation are incorrect.*

**Response:**

We agree that factors like salinity, nutrients, prey availability, turbidity, light, and the species' depth habitat are important in controlling their distribution. Our habitat depth inference is based on previous studies using multi plankton net samples from the Northern Indian Ocean (lines 379–381; Fig. S5), which suggest that *G. bulloides* thrives at 0–50 m and *N. dutertrei* at 50–100 m.

We did not intend to suggest that the relationship between chlorophyll-a and planktic foraminifera applies universally to all species. Rather, this relationship specifically applies to *G. bulloides* and *N. dutertrei* in the region of the Bay of Bengal. We also discussed the

limitations of this proxy, such as the effects of suspended sediment and upwelling on the depth of chlorophyll-a maxima (lines 389–412).

In the revision, we will the update lines 378–381 with a more detailed explanation, as follows:

"The premise behind our analysis on the abundance of two planktic foraminiferal species, *Globigerina bulloides* and *Neogloboquadrina dutertrei*, which exhibit optimal thriving conditions at water depths of 0-50 meters and 50-100 meters, respectively, based on multi-plankton net samples from the northern Indian Ocean (Tapia et al., 2022) (Fig. S5). The oxygen isotope-based apparent calcification depths of these species further indicate shallow water habitat for *G. bulloides* and deeper water habitat for *N. dutertrei* (Stainbank et al., 2019). Given that the salinity and temperature of both shallow and deeper waters in the BoB align with these species' ecological niches, the Chl a concentrations regulates their populations (Lombard et al., 2009; Bijma et al., 1990; Munir et al., 2022; Kuroyanagi and Kawahata, 2004; Maeda et al., 2022)."

**Comment 3:** *2. The authors calculated relationship between SST and ice core $CO_2$ concentration/solar insolation at 30° These R-sq values explain 54% and 8% variability of internal and external earth system forcing, respectively (Line 354-355). However, the authors had big problem on statistics. A correlation between two variables cannot be interpreted as they mentioned. These data should be performed by factor analysis, EOF or principal component analysis. Moreover, in the Results and Discussion 3.1, 19 samples were analyzed for reconstruction of past SST variations. The authors calculated an average SST at a specific time window. It makes no sense that an average temperature with 1 STD was calculated by only 3-4 data at the time window.*

**Response:**

We conducted a correlation analysis to assess the sensitivity between SST and ice core $CO_2$ concentrations/solar insolation. This is standard practice for examining numerical relationships between these variables (https://doi.org/10.1175/2011JCLI4078.1).

The referee's suggestion to apply factor analysis, EOF, or PCA may be more suited for multivariate datasets. However, we find it unclear what exactly the referee is proposing without further clarification. Applying PCA to just three-time series (SST, $CO_2$ & insolation) would not yield meaningful statistical significance.

As for the averaging of SST, the mean values and standard deviations were calculated using all available data points within each time window. Although some windows contained only 3-4 data points, the averages and standard deviations remain statistically valid for capturing the central tendency and variability. While a larger sample size would offer stronger statistical power, this constraint is inherent to paleoclimate data.

**Comment 4:** *3. Line 180-183: The authors mentioned that a strong coherence of $\delta^{18}O$ variability in ruber was observed in multiple sites adjacent to our core location (Rashid et al., 2011; Govil and Divakar Naidu, 2011; Clemens et al., 2021) confirming the proposed age-depth model. A good age model is dependent on how precise of radiocarbon dates, not for similar variations of adjacent cores. Some discrepancies can be found among these stable oxygen isotope records (Figure S3).*

**Response:**

Our age-depth model is constructed from 7 radiocarbon ages and interpolated for each depth strata using Bayesian statistics. The observed strong coherence of $\delta^{18}O$ variability in *G. ruber* between our core site and adjacent sites supports the validity of our age-depth model. We will revise lines 180-183 to read as follows:

"A strong coherence of $\delta^{18}O$ variability in *G. ruber* was observed across multiple adjacent sites (Rashid et al., 2011; Govil and Divakar Naidu, 2011; Clemens et al., 2021), which substantiates the proposed age-depth model (Fig. S3)."

The discrepancies observed in the $\delta^{18}O$ records from multiple sites can be attributed to inter-site hydrographic variations. Nevertheless, the overall temporal trends are consistently recorded across all sites.

**Comment 5:** *4. Introduction, method and results of the MS are jumbled up. Core information and results of age model have too many redundancies. The information can be found in several paragraphs.*

**Response:**

We did not understand the referee's comments on the manuscript structure. We would appreciate it if the referee could provide more explicit feedback on this issue.

**Comment 6:** *5. Discussions with multiple SST records are unclear.*

**Response:**

The discussion addresses the discrepancies in SST differences between the Late Holocene and the LGM as reported by different temperature proxies from BoB. We have examined the reasons for these discrepancies in lines 329–351.

**Comment 7:** *6. Paragraph of 2. Materials and Methods is too long.*

**Response:** Our materials and methods section is as streamlined as possible.

**Comment 8:** *7. Technical aspects: Figure 1: surface currents cannot be excluded.*

**Response:**

Thanks! We have included surface currents in the revised Figure 1 (attached below).

"

[Figure]

**Figure 1: Site map and climatology.** a) modern-day climatology (1998 – 2019) showing the distribution of rainfall minus evaporation (mm/day), wind vectors (black arrow), and surface ocean circulation (green dashed arrow) during the period of SASM (June, July, August, and September) (Kalnay et al., 1996; Schneider et al., 2013; Phillips et al., 2021). The study site, MGS17/GC02 (depicted as a filled orange diamond with a black boundary), along with other locations (represented by light yellow-filled circles with black boundaries), which are discussed in the text; b) contours of mass-corrected vertically integrated moisture flux during the SASM for the period of 1979 - 2017 (Trenberth and Fasullo, 2022), superimposed with 48-hour HYSPLIT backward air mass trajectories (Stein et al., 2015) at multiple altitudes (200 m, 500 m, and 1000 m above mean sea level) for the days prior to SASM precipitation events in 2023 at Kolkata. c) Plot of monthly Sea Surface Salinity (Reagan et al., 2024) and moisture flux (Trenberth and Fasullo, 2022) distribution over the study site (MGS17/GC02) obtained from world ocean atlas climatology resolved at 4°x4° grid space (indicated by the red dashed square in panel b); d) Long-term average monthly river discharge of Ganges and Brahmaputra to BoB (Jana et al., 2015)."

**Comment 9:** *8. Others: Abbreviations: Bay of Bengal (BoB), sea surface temperature (SST), kilo years (kys)*

**Response:** Thanks! We will address the abbreviation issue in the revised manuscript.

**Reference:**

Berggren, W. A.: Carbon cycling in the glacial ocean: Constraints on the ocean's role in global change, 80–82 pp., https://doi.org/10.1016/0012-8252(95)90055-1, 1995.

Bijma, J., Faber, W. W., and Hemleben, C.: Temperature and salinity limits for growth and survival of some planktonic foraminifers in laboratory cultures, J. Foraminifer. Res., 20, 95–116, https://doi.org/https://doi.org/10.2113/gsjfr.20.2.95, 1990.

Clemens, S. C., Yamamoto, M., Thirumalai, K., Giosan, L., Richey, J. N., Nilsson-Kerr, K., Rosenthal, Y., Anand, P., and McGrath, S. M.: Remote and local drivers of pleistocene South Asian summer monsoon precipitation: A test for future predictions, Sci. Adv., 7, 1–16, https://doi.org/10.1126/sciadv.abg3848, 2021.

Govil, P. and Divakar Naidu, P.: Variations of Indian monsoon precipitation during the last 32kyr reflected in the surface hydrography of the Western Bay of Bengal, Quat. Sci. Rev., 30,

3871–3879, https://doi.org/10.1016/j.quascirev.2011.10.004, 2011.

Jana, S., Gangopadhyay, A., and Chakraborty, A.: Impact of seasonal river input on the Bay of Bengal simulation, Cont. Shelf Res., https://doi.org/10.1016/j.csr.2015.05.001, 2015.

Kalnay, E., Kanamitsu, M., Kistler, R., Collins, W., Deaven, D., Gandin, L., Iredell, M., Saha, S., White, G., Woollen, J., Zhu, Y., Chelliah, M., Ebisuzaki, W., Higgins, W., Janowiak, J., Mo, K. C., Ropelewski, C., Wang, J., Leetmaa, A., Reynolds, R., Jenne, R., and Joseph, D.: The NCEP/NCAR 40-year reanalysis project, Bull. Amer. Meteor. Soc., 77, 437–470, 1996.

Kuroyanagi, A. and Kawahata, H.: Vertical distribution of living planktonic foraminifera in the seas around Japan, Mar. Micropaleontol., 53, 173–196, https://doi.org/10.1016/j.marmicro.2004.06.001, 2004.

Lombard, F., Labeyrie, L., Michel, E., Spero, H. J., and Lea, D. W.: Modelling the temperature dependent growth rates of planktic foraminifera, Mar. Micropaleontol., 70, 1–7, https://doi.org/10.1016/j.marmicro.2008.09.004, 2009.

Maeda, A., Kuroyanagi, A., Iguchi, A., Gaye, B., Rixen, T., Nishi, H., and Kawahata, H.: Seasonal variation of fluxes of planktic foraminiferal tests collected by a time-series sediment trap in the central Bay of Bengal during three different years, Deep. Res. Part I Oceanogr. Res. Pap., 183, 103718, https://doi.org/10.1016/j.dsr.2022.103718, 2022.

Munir, S., Sun, J., Morton, S. L., Zhang, X., and Ding, C.: Horizontal Distribution and Carbon Biomass of Planktonic Foraminifera in the Eastern Indian Ocean, 14, 1–15, https://doi.org/10.3390/w14132048, 2022.

Phillips, H. E., Tandon, A., Furue, R., Hood, R., Ummenhofer, C. C., Benthuysen, J. A., Menezes, V., Hu, S., Webber, B., Sanchez-Franks, A., Cherian, D., Shroyer, E., Feng, M., Wijesekera, H., Chatterjee, A., Yu, L., Hermes, J., Murtugudde, R., Tozuka, T., Su, D., Singh, A., Centurioni, L., Prakash, S., and Wiggert, J.: Progress in understanding of Indian Ocean circulation, variability, air-sea exchange, and impacts on biogeochemistry, Ocean Sci., 17, 1677–1751, https://doi.org/10.5194/os-17-1677-2021, 2021.

Rashid, H., England, E., Thompson, L., and Polyak, L.: Late glacial to holocene indian summer monsoon variability based upon sediment records taken from the bay of Bengal, Terr. Atmos. Ocean. Sci., https://doi.org/10.3319/TAO.2010.09.17.02(TibXS), 2011.

Reagan, J. R., Boyer, T. P., García, H. E., Locarnini, R. A., Baranova, O. K., Bouchard, C., Cross, S. L., Mishonov, A. V., Paver, C. R., Seidov, D., Wang, Z., and Dukhovskoy, D.: World Ocean Atlas 2023., NOAA Natl. Centers Environ. Information.https//www.ncei.noaa.gov/archive/accession/0270533. Accessed [25/04/2024], 2024.

Schneider, D. P., Deser, C., J. Fasullo, and Trenberth, K. E.: Climate Data Guide Spurs Discovery and Understanding., Eos Trans. AGU, 94, 121–122, https://doi.org/https://doi.org/10.1002/2013eo130001, 2013.

Stainbank, S., Kroon, D., Rüggeberg, A., Raddatz, J., De Leau, E. S., Zhang, M., and Spezzaferri, S.: Controls on planktonic foraminifera apparent calcification depths for the northern equatorial Indian Ocean, 1–34 pp., https://doi.org/10.1371/journal.pone.0222299, 2019.

Stein, A. F., Draxler, R. ., Rolph, G. D., Stunder, B. J. B., Cohen, M. D., and Ngan, F.: NOAA's HYSPLIT atmospheric transport and dispersion modeling system, Bull. Amer. Meteor. Soc., 96, 2059–2077, https://doi.org/http://dx.doi.org/10.1175/BAMS-D-14-00110.1,

2015.

Tapia, R., Ho, S. L., Wang, H. Y., Groeneveld, J., and Mohtadi, M.: Contrasting vertical distributions of recent planktic foraminifera off Indonesia during the southeast monsoon: implications for paleoceanographic reconstructions, 19, 3185–3208, https://doi.org/10.5194/bg-19-3185-2022, 2022.

Trenberth, K. and Fasullo, J.: "The Climate Data Guide: ERA-Interim: derived components." Retrieved from https://climatedataguide.ucar.edu/climate-data/era-interim-derived-components on 2024-03-30., Natl. Cent. Atmos. Res. Staff (Eds). Last Modif. 2022-11-07, 2022.